# Association between physical activity and changes in intestinal microbiota composition: A systematic review

**Viviana Aya[1], Alberto Flórez[2], Luis Perez[1], Juan David Ramírez**[ORCID][1] *

**1** Centro de Investigaciones en Microbiología y Biotecnología-UR (CIMBIUR), Facultad de Ciencias Naturales, Universidad del Rosario, Bogotá, Colombia, **2** Grupo In-Novum Educatio, Facultad de Educación, Pontificia Universidad Javeriana, Bogotá, Colombia

* juand.ramirez@urosario.edu.co

## Abstract

### Introduction

The intestinal microbiota comprises bacteria, fungi, archaea, protists, helminths and viruses that symbiotically inhabit the digestive system. To date, research has provided limited data on the possible association between an active lifestyle and a healthy composition of human microbiota. This review was aimed to summarize the results of human studies comparing the microbiome of healthy individuals with different physical activity amounts.

### Methods

We searched Medline/Ovid, NIH/PubMed, and Academic Search Complete between August–October 2020. Inclusion criteria comprised: (a) cross-sectional studies focused on comparing gut microbiome among subjects with different physical activity levels; (b) studies describing human gut microbiome responses to any type of exercise stimulus; (c) studies containing healthy adult women and men. We excluded studies containing diet modifications, probiotic or prebiotic consumption, as well as studies focused on diabetes, hypertension, cancer, hormonal dysfunction. Methodological quality and risk of bias for each study were assessed using the Risk Of Bias In Non-randomized Studies—of Interventions tool. The results from cross-sectional and longitudinal studies are shown independently.

### Results

A total of 17 articles were eligible for inclusion: ten cross-sectional and seven longitudinal studies. Main outcomes vary significantly according to physical activity amounts in longitudinal studies. We identified discrete changes in diversity indexes and relative abundance of certain bacteria in active people.

### Conclusion

As literature in this field is rapidly growing, it is important that studies incorporate diverse methods to evaluate other aspects related to active lifestyles such as sleep and dietary

**Data Availability Statement:** All relevant data are within the manuscript and its Supporting information files.

**Funding:** The author(s) received no specific funding for this work.

**Competing interests:** The authors have declared that no competing interests exist.

patterns. Exploration of other groups such as viruses, archaea and parasites may lead to a better understanding of gut microbiota adaptation to physical activity and sports and its potentially beneficial effects on host metabolism and endurance.

## Introduction

The intestinal microbiota comprises bacteria, fungi, archaea, protists, helminths and viruses that symbiotically inhabit the human digestive system, with five bacterium phyla—*Firmicutes*, *Bacteroidetes*, *Actinobacteria*, *Proteobacteria*, and *Verrucomicrobia*—representing the predominant microorganisms in the gut [1, 2]. The term "microbiome" refers to the collective genome of these microbes [1, 2].

Despite coordinated efforts by several international consortia to define the composition of a healthy microbiota [3], the term currently remains incomplete given the many intrinsic and extrinsic factors associated with gut ecosystems [4–7].

Numerous studies have aimed to establish the role of environmental and behavioral factors on microbiota composition [8–14]. Diet is currently considered the main extrinsic factor [15, 16] followed by sleep, circadian rhythm [17], and physical activity [7]. Similarly, the presence and progression of disease might change the abundance and diversity of bacteria [4]: chronic illnesses such as irritable bowel syndrome (IBS) [8, 9], type 2 diabetes (T2D) [10], hypertension [11], and cancer [12] are associated with abnormal microbiota composition and function. Changes in microbiota diversity may diminish the abundance of beneficial bacteria while favoring the growth of potentially pathogenic microorganisms, a process known as "dysbiosis", which can further impact host metabolism [18, 19]. In obesity, for example, changes in the abundance of Bacteroides and Firmicutes (B/F ratio), may promote fat storage, increase energy collection from nutrients, and decrease energy expenditure [13]. The stability and diversity of microbiota also vary across age [14, 20] with larger microorganism diversity associated with adulthood with healthy habits [21, 22].

Active lifestyle behaviors improve several metabolic and inflammatory parameters in chronic diseases: exercise regimes have been used as therapeutic strategies against obesity and T2D [23]. Physical activity promotes adaptational changes on human metabolic capacities to reach a specific goal, which could be competitive in the case of athletes, or recreational and aesthetic in the non-competitive population. Diet is also a major target to meet these objectives as the consumption of dietary supplements is common in active people [24, 25] and probiotic consumption emerges as a profitable market due to its possible effects on gut epithelium homeostasis, especially for competing athletes [26]. However, this is a growing research field and there are still some concerns about its positive impact on human gut microbiota [27].

It is still unclear whether an active lifestyle, a healthy diet, or a combination of both can influence intestinal microbiota towards a healthy state. Animal models have allowed researchers to develop physiological and biochemical protocols [28–38] to explore the functional effects of exercise on the microbiome [37, 39, 40], and cross-sectional and longitudinal studies have tried to describe the effects of physical activity on the microbiome composition of active versus non-active adult humans. The heterogeneity of methodological approaches and the lack of standardized criteria for active/non-active people is one of the largest challenges in this research field.

This review aims to summarize the results of all human studies comparing the microbiome composition of healthy individuals with different physical activity amounts (PAA).

## Methods

### Reporting

Results from this study were reported based on the Preferred Reporting Items for Systematic Reviews and Meta-Analyses (PRISMA) statement guidelines [31].

### Search strategy

A computerized search was conducted between August–October 2020 using standardized English search terms assigned by the MeSH and EMTRE indexes using the Boolean operators OR/AND: "exercise" OR "physical activity" AND "human" AND "gastrointestinal microbiome" OR "gut microbiota". The databases consulted included Medline/Ovid, NIH/PubMed, and Academic Search Complete. The results from cross-sectional and longitudinal studies are shown independently.

### Inclusion and exclusion criteria

We included the following research in our review: (a) cross-sectional studies focused on comparing gut microbiome among subjects with different physical activity levels—from athletes to inactive individuals—, using guidelines from the American College of Sports Medicine (ACSM) [41]; (b) studies describing human gut microbiome responses to any type of exercise stimulus; (c) studies containing healthy adult women and men (18–45 years old); (d) studies written in English.

We excluded studies containing diet modifications, probiotic or prebiotic consumption, as well as studies focused on diabetes, hypertension, cancer, hormonal dysfunction, or related illnesses since evidence suggests that these conditions may lead to significant changes in the composition of gut microbiota. Reviews, comments, letters, interviews, and book chapters were also excluded. PRISMA flow diagram (Fig 1) shows the screening process for this systematic review [42].

### Quality assessment

Methodological quality and risk of bias for each study were assessed using the Risk Of Bias In Non-randomized Studies—of Interventions tool (ROBINS-I) [43]. This tool provides a detailed framework for assessing the risk of bias domains from Non-randomized studies of interventions (NRSIs).

Once a target trial specificity to the study was designed and confounding domains were listed, the risk of bias was assessed specifically for the comparisons of interest to this review. The overall risk of bias judgment can be found in S1 Table.

Supplementary document includes a checklist based on the Preferred Reporting Items for Systematic Reviews and Meta Analyses (PRISMA).

## Results

### Literature search

654 articles were retrieved from the databases. Duplicate studies were identified and removed, leaving only 467 articles for screening. Once records were screened by title and abstract, a total of 359 articles were excluded. After careful reading of the methodology section of the remaining 108 potential eligible articles, exclusion criteria were applied.

Finally, a total of 17 studies were included in this review. Recorded outcome measures included differences for α and β diversity and relative abundance ($p < 0.05$). Transcriptional

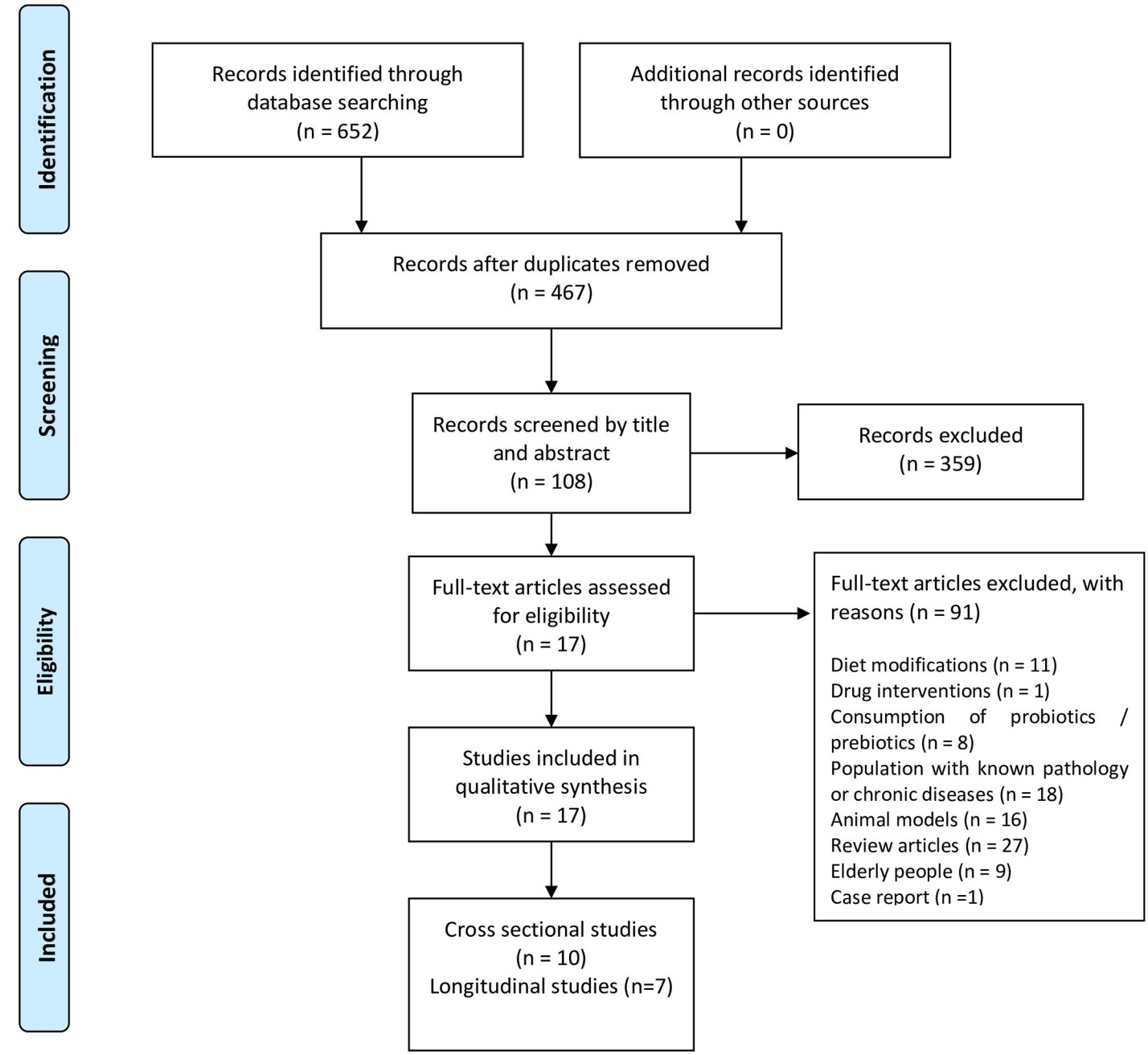

**Fig 1. PRISMA flow diagram employed for this systematic review.**

and metabolomic data extracted from feces were included. Measurements or description of physical activity amounts (PAA) were used to classify results based on inactive, active, and athletic subjects.

## Different levels of physical activity on gut microbiota

Table 1 shows a set of studies aiming to establish whether meeting the recommended physical activity quotas [41] influences microbiota composition measured as physical activity amounts (PAA) and abundance/diversity respectively.

**Table 1. Summary of cross-sectional studies comparing intestinal microbiota between groups with different Physical Activity Amounts (PAA).**

| Reference | Publication year | N | Sample | Comparison axis | Results | Country |
|---|---|---|---|---|---|---|
| ** Clarke et al. [44] Barton et al. [45] | 2014 | 86 | Rugby players (29 ± 4) and two control groups with different BMI (29 ±6) | Athletes and non-athletes with distinct BMI ≤25 ->28 | Higher diversity in athletes (Shannon index, p = 0.0064) Relative abundance Athletes vs IMC ≤25 ↑ 40 taxa ↓ *Bacteriodetes* ↓ *Lactobacillaceae* ↓ *Lactobacillus* | Ireland |
| | | | | | **Athletes vs IMC >28** ↑ 48 taxa ↑*Akkermansiaceae (family)* ↑*Akkermansiaceae (genus)* ↓*Bacteriodetes (genus)* | |
| Estaki et al. [46] | 2016 | 39 | Healthy young men and women (26.2 ± 5.5) | Categories of cardiorespiratory fitness: High–Avg–Low | No differences in α and β diversity | Canada |
| Bressa et al. [47] | 2017 | 40 | Middle-aged women (ACT 30.7 ± 5.9 –SED 32.2 ± 8.7) | Physical activity level and sedentary behavior | No differences for α and β diversity among groups **Sedentary women** ↑*Barnesiellaceae* (family & genus) ↑*Odoribacteraceae* (family & genus) ↑*Bifidobacterium* (genus) ↑*Turicibacter* (genus) ↑*Clostridiales* (genus), ↑*Coprococcus* (genus) ↑*Ruminococcus* (genus) **Women with an active lifestyle** ↑*Faecalibacterium prausnitzii* (spp.) ↑ *Roseburia hominis* (spp.) ↑ *Akkermansia muciniphila* (spp.) | Spain |
| Petersen et al. [48] | 2017 | 71 | Cyclists (women and men) with ≥2 years participating in competitive events | Performance level: professional vs amateurs | Higher diversity in cluster 3 (11 professional and 3 amateur cyclists) Shannon index p = 0.0004 Higher abundance of the genus: *Bacteroides, Prevotella, Eubacterium, Ruminococcus,* and *Akkermansia* | United States |
| Yang et al. [49] | 2017 | 71 | Premenopausal women age between 19 and 49 years | Cardiorespiratory fitness (CRF): High–Low | High CRF ↑*Bacteroides* ↓ *Eubacterium rectale* | Finland |
| Whisner et al. [50] | 2018 | 82 | University students (men and women) (18.4 ± 0.6) | Physical activity level and sedentary behavior | No differences in α and β diversity | United States |
| Durk et al. [51] | 2019 | 38 | Apparently healthy men and women (25.7 ± 2.2) | Comparison between gender and oxygen consumption (VO2peak) | No differences in α and β diversity | United States |
| Jang et al. [52] | 2019 | 45 | Bodybuilding (n = 15), athletes (n = 15), non-athlete control group (n = 15) | Differences among sporting activity | Relative abundance in bodybuilders (p < 0.05) ↑*Faecalibacterium* ↑*Sutterella* ↑*Clostridium* ↑*Haemophilus* ↑*Eisenbergiella* ↓*Bifidobacterium* ↓*Parasutterella* Athletes and control group (p < 0.05) ↑*Bifidobacterium adolescentis* ↑*Bifidobacterium longum* ↑*Lactobacillus sakei* ↑*Blautia wexlerae* ↑*Eubacterium hallii* | South Korea |
| O'Donovan et al. [53] | 2019 | 37 | International level athletes | Differences among sports classification groups | ↑*Eubacterium rectale* ↑*Polynucleobacter necessarius* ↑*Faecalibacterium prausnitzii* ↑*Bacteroides vulgatus* ↑*Gordonibacter massiliensis* | Ireland |

(*Continued*)

**Table 1.** (Continued)

| Reference | Publication year | N | Sample | Comparison axis | Results | Country |
|---|---|---|---|---|---|---|
| Liang et al. [54] | 2019 | 28 | Wushu martial arts athletes (20.1 ± 1.8) | High (H) and Low (L) levels of competition | H group<br>Higher α diversity (Shannon index p = 0.019 and Simpson diversity index p = 0.001)<br>↑*Parabacteroides*, ↑*Phascolarctobacterium*<br>↑*Oscillibacter* ↑*Bilophila* ↓*Megasphaera* | China |

* Reported findings refer to the composition of the microbiota in terms of diversity (α and β) and species abundance where arrows denote increase (↑) or decrease (↓).

Only significant results are shown (p < 0.05)

**CRF** = cardiorespiratory fitness

** Studies including same population.

Notable differences have been described between competing athletes and inactive people: (a) greater microbiota α-diversity has been reported in athletes—highly associated with dietary patterns and protein consumption [44, 45]; and (b) a significative abundance of *Lachnospiraceae*, *Akkermansiaceae* and *Faecalibacterium* bacteria coupled with a lower abundance in *Bacteroidetes* phylum has been reported in active women [47].

Researchers have also considered associations between cardiorespiratory fitness (CRF) and the composition of gut microbiota, although no significant differences in α or β diversity indexes were reported between high, medium, and low VO2 consumption [46, 49, 51]. VO2peak was a significant predictor of α-diversity, with the Species Richness index significantly (p = 0.011) associated with increasing VO2peak (Radj2 = 0.204) [46]. It is important to note that no differences related to gender and oxygen consumption have been reported in the gut microbiota field [51].

## Gut microbiota composition in studies involving athletes

We included studies that compared microbiota composition and diversity of individuals from different sporting disciplines [45, 48, 52–55]: (a) some disciplines were strongly associated with a relative abundance of bacteria as described in Table 1 [52]; (b) athletes from distinct disciplines and level of competition displayed significative differences in microbiota diversity and species richness [48, 54]; and (c) high-performing individuals have been reported with a greater abundance of the genera *Parabacteroides*, *Phascolarctobacterium*, *Oscillibacter*, *Bilophila* and a lower abundance of *Megasphaera* [54].

Characteristics related to training loads were also explored. A study positively correlated α and β-diversity with decreased training volume per week during a two-week follow-up on a group of swimmers [56]. In a different study, dynamic and static components of sports practice [57] were used to cluster a sample of Olympic athletes revealing no significant differences in diversity indexes in groups categorized by energy demand, with the whole sample exhibiting an abundance of the species *Eubacterium rectale*, *Polynucleobacter necessarius*, *Faecalibacterium prausnitzii*, *Bacteroides vulgatus* and *Gordonibacter massiliensis* [53].

## Changes in composition and function of the intestinal microbiota after an exercise program and sporting events

Fig 2 shows how phenotypic characteristics of the host and stimulus exposure times can induce changes in specific groups of bacteria when starting an exercise program. Body mass index (BMI) appears to be a determining factor in microbiota response to exercise. Fecal microbiota

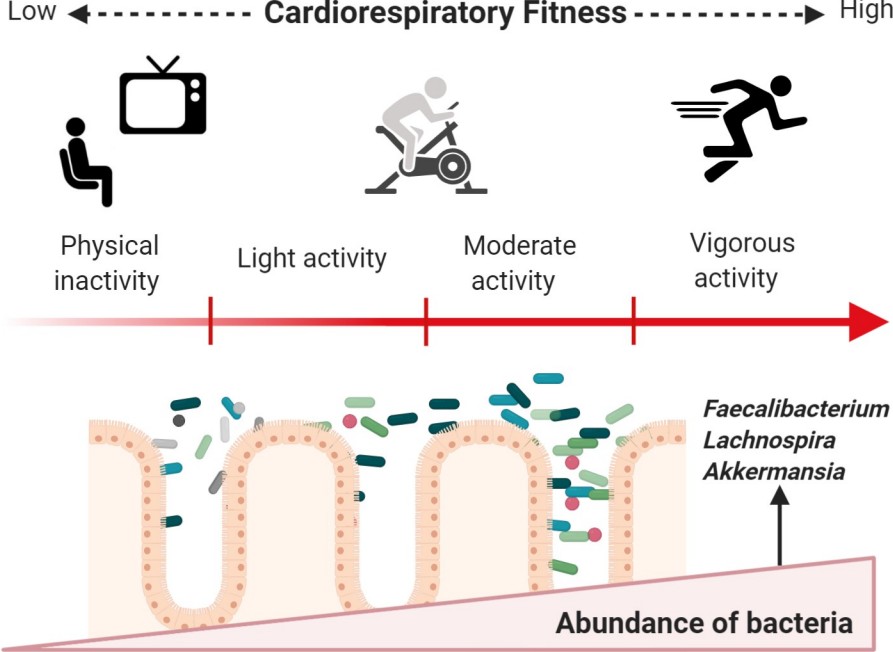

**Progressive increase of physical activity level generates changes in the intestinal microbiota**

**Fig 2. Exercise induces changes in gut microbiota through enhanced CRF in previously inactive subjects.** Once a subject increases PAA, a series of beneficial molecular adaptations are induced allowing the enhancement of CRF. Major oxygen consumption is related to lower cardiometabolic risk, which may occur by progressively increasing energy-demanding activities based on endurance training. Physiological modifications occur, and the gut microbiota does not appear to play a role in this process. Recent research has provided insight into a progressive increase of helpful members from different phyla of bacteria. However, these changes could depend on BMI status, energy demand, and exposure time to exercise. Figure created with Biorender.com.

from apparently healthy individuals with a BMI $\geq$ 25 kg/m$^2$ shows discrete incremental changes regarding relative abundance of *Actinobacteria*, *Bacteroides*, *Firmicutes*, *Proteobacteria*, and *Verrucomicrobia* phyla after 6 weeks of supervised aerobic training (Table 2). Gut microbiota from lean subjects responds to aerobic exercise by increasing the abundance of species from *Faecalibacterium* spp. and *Lachnospira* spp., and by reducing *Bacteroides* members [58, 59]. The only randomized intervention study to date reported a significant difference in Shannon index values with higher diversity in groups after 3 and 6 months of vigorous-intensity physical activity (70% of peak VO2) in subjects aged between 20 and 40 years who were overweight or obese in comparison to control group. Significant reduction of fat mass and increased CRF were observed in both groups while dietary values remained similar before and after the intervention [60]. To date, only one study has reported microbiota findings related to high-intensity interval training (HIIT) [61]; the authors of this non-randomized trial observed an increased in the abundance of *Subdoligranulumwa* (genus) in lean men (p = 0.0037) after three weeks of cycloergometer workout.

To identify whether there are changes after an endurance event, two studies collected samples before and after both a half-marathon and a marathon. The athletes had similar characteristics in terms of body composition, training level, diet, and age. The most significant change

**Table 2. Longitudinal studies containing two or more samples of feces, before and after exercise activities.**

| Reference | Study type** | Year | Sample | Observation/intervention time | Comparison axis | Findings* | Country |
|---|---|---|---|---|---|---|---|
| Allen et al. [58] | Intervention | 2018 | N = 32 Women and men with different BMI: Lean n = 18 (X̄ 25 years old) Obese n = 14 (X̄ 31 years old) | Six weeks of aerobic exercise, duration 30 to 60 min and moderate-high intensity (60–75% HR) | Before and after exercise intervention | LEAN ↓*Bacteroides* ↑*Faecalibacterium* spp. ↑*Lachnospira* spp.<br><br>OBESE ↓*Faecalibacterium* spp. ↑*Bacteroides* ↑*Colinsella* | United States |
| Munukka et al. [59] | Intervention | 2018 | N = 17 Sedentary middle age women BMI >27.5 kg/m2. | Six weeks of aerobic exercise | Before and after exercise intervention | ↑*Dorea* ↑*Anaerofilum* ↑*Akkermansia* ↓*Porphyromonadaceae* ↓*Odoribacter* ↓*Desulfovibrionaceae* ↓*Enterobacteriaceae* | Finland |
| Kern et al. [60] | Intervention | 2020 | N = 88 Overweight and obese people age between 20–45 years old | Six weeks of intervention with different types of activities CON: No exercise (n = 14) BIKE: Active transport (n = 19) MOD: leisure time exercise (n = 31) VIG: vigorous supervised exercise (n = 24) | Control group vs rest of groups | Three months intervention VIG α diversity p = 0.012 6 months intervention VIG α diversity p = 0.059 | Denmark |
| Zhao et al. [62] | Observational | 2018 | N = 20 Amateur athletes (X̄ 31 years old) | Gut microbiota before and after running a half marathon, distance → 21 km BEF: Feces analyzed before half marathon AFT: Feces analyzed after half marathon | Before and After half marathon | BEF: ↓*Bacteroides coprophilus* AFT: ↑*Pseudobutyrivibrio* ↑*Coprococcus_2* ↑*Mitsuokella* | China |
| Scheiman et al. [55] | Observational | 2019 | N = 15 Professional marathonists (X̄ 27.4 years old) | Feces recollected during one week before and after the Boston marathon, distance → 42 km | Before and After marathon | ↑ *Veillonella* genus | United States |
| Hampton-Marcell et al. [56] | Observational | 2020 | N = 13 University swimmers aged between 18 and 24 years old | Feces recollected in three temporal phases of two weeks | Between phases and volume training | No differences for α—β diversity and abundance | United States |
| Rettedal et al. [61] | Intervention | 2020 | N = 29 men aged 20–45 years old (n = 14 lean; n = 15 overweihtg) | Nine sessions of HIIT on non-consecutive days | Pre–post HIIT intervention | No differences for α—β diversity and abundance between samples or exercise intervention. | New Zealand |

*The reported findings refer to the composition of the microbiome in terms of diversity (α and β) and species abundance. Arrows denote an increase (↑) or decrease (↓). Significant results are shown (p < 0.05).

** Types of studies included: Intervention: The selected sample carried out an exercise program for a certain period. Observational: The selected sample was followed before and after a sporting event. **HR** = heart rate; **HIIT** = high intensity interval training.

in microbiome composition was relative abundance. In the case of amateur athletes who ran 21 km, the presence of *Pseudobutyrivibrio*, *Coprococcus_2*, *Collinsella*, and *Mitsuokella* were significantly greater at the end of the race [62].

Another study included repeated samples from professionals athletes 1 week before and after running the Boston Marathon [55]. The results revealed a significant increase in *Veillonella*, a Gram-negative, anaerobic bacteria commonly found in gut and oral microbiota—with unique physiology including the capacity to obtain energy through lactate fermentation and

their inability to utilize glucose [55]. Metagenomic sequencing revealed an overrepresentation in the methylmalonyl-CoA pathway [55]. Isolation and subsequent treatment in mice with the strain *Veillonella atypica* were carried out to test whether mice inoculated with this bacterium or with *Lactobacillus bulgaricus* (control) exhibited altered responses after endurance training. The results revealed that animals treated with *V. atypica* showed a significant reduction in post-training proinflammatory cytokine levels, as well as better performance. No changes in GLUT4 glucose transporters were observed. To determine whether the richness of *Veillonella* causes functional changes in the emission of messengers such as short-chain fatty acids (SCFA), propionate from three samples was directly extracted and quantified using mass spectrometry. The results revealed an increased abundance of *Veillonella* and an improvement in lactate pathways. However, subsequent tests in animal models failed to demonstrate that resulting lactate could cross the lumen barrier and impact other tissues, such as muscle, brain or liver. Still, it could cross the barrier to the intestinal lumen [55].

## Physical activity substantially improves metabolite synthesis associated with intestinal microbiota

Exercise in physically inactive individuals produces changes in the composition of the microbiome and improves the synthesis of metabolites associated with the intestinal microbiota. Table 3 describes findings related to functional changes in gut microbiota, where diverse techniques and predictive approaches were used across studies; Allen et al. [58] used gas chromatography to quantify Short Chain Fatty Acids (SCFAs) from fecal samples of people who started a 6-week program of mostly aerobic exercise. Participants were differentiated by BMI (lean and obese). Only samples from lean subjects (BMI <25.0 kg/m$^2$) exhibited a significant increase in SCFAs concentration after the training period, whereas the concentrations of acetate, propionate, and butyrate in the obese group (BMI >30 kg/m$^2$) remained unchanged. The significant increases in butyrate-producing bacteria in lean people such as *Roseburia* spp., *Lachnospira* spp., *Lachnospriaceae*, *Clostridiales*, and *Faecalibacterium* (Fig 2) were positively correlated to changes in butyrate concentrations and butyryl-CoA: acetate CoA-transferase (BCoAT) genes.

Performing energy-demanding activities (e.g., a half marathon) has also been associated with modifications in metabolites related to gut microorganisms. One study using metabolomic analysis with liquid chromatography of samples before and after a half marathon event in Shanghai reported an increase in the concentration of approximately 40 metabolites, mainly organic acids [62]. Results also revealed a decrease in 19 compounds (fold change > 0). The metabolic routes with the greatest changes after the race were pentose phosphate more enriched with a value of q = 0.0071, while biosynthesis of phenylalanine, tyrosine, and tryptophan was highly decreased [62]. In a similar study, Scheiman and collaborators used meta-omics analysis in order to elucidate the metabolic contribution of *Veillonella* species, findings provide and insight about the role of bacteria in the systemic degradation of lactic acid after a high performance activity [55].

## Discussion

Several microorganisms found in the gastrointestinal system of active individuals and elite athletes classify as beneficial bacteria (Fig 3). Tables 1 and 2 show an increased abundance of butyrate-producing bacteria like *Eubacterium rectale* in competitive athletes, which predominantly uses dietary starch but can also utilize by-products of resistant starch (RS) degradation produced by other bacteria [63, 64]. However, a greater abundance of this species has also been found in obese people and is linked to inflammatory status and dysbiosis [65]. Table 3 shows

**Table 3. Description of possible functional changes associated with gut microbiota in physical activity and sports activities.**

| Reference | Metabolic approach used across studies | Outcomes related to functional changes | Possible pathways related to gut microbiota function and composition |
|---|---|---|---|
| Barton et al. [45] | Metagenomic analysis of fecal samples from rugby players (n = 40) and controls (n = 46) Quantification of SCFAs levels in feces gas chromatography–mass spectrometry. | Athletes had highest mean abundance across 29 of the 34 metabolic pathways categories stablished by authors. Levels of SCFAs were significantly higher in the athlete's group: acetate (p<0.001), propionate (p<0.001), butyrate (p<0.001) and valerate (p = 0.011). | Carbohydrate degradation Production of secondary metabolites and cofactors Production of SCFAs |
| Estaki et al. [46] | Analysis of SCFAs from the feces by gas chromatography (N = 39 healthy young adults with different levels of CRF) | VO2peak was strongly correlated with butyric acid mainly across HI and AVG fitness participants. Propionic and acetic acid were inversely correlated to VO2peak and were represented across LOW fitness participants. | The abundance of key butyrate-producing members from *Clostridiales*, *Roseburia*, *Lachnospiraceae*, and *Erysipelotrichaceae* genera was associated with the production of butyric acid. |
| Bressa et al. [47] | Fecal enzymatic activity from feces was quantified using a semi quantitative method (active and sedentary middle age women, N = 40) | The activity of cysteine aminopeptidase in feces was significantly higher in the active group than in the sedentary group. | Cysteine aminopeptidase activity was negatively correlated with the presence of *Bacteroides*. |
| Petersen et al. [48] | Alignment of mWGS reads to the KEGG database (N = 33 cyclists) | *Prevotella* transcriptional activity was positively correlated to three KEGG pathways and negatively correlated to two amino acid metabolism pathways. Positive associations between *Methanobrevibacter* and methane metabolism were reported. | Upregulation of methane metabolism was related to citrate cycle, oxidative phosphorylation, and pyruvate metabolism. Similarly, pathways in SCFA production, propanoate metabolism and butanoate metabolism, were upregulated along with methane metabolism. |
| O' Donovan et al. [53] | Fecal samples underwent 1H-NMR and UPLC-MS analysis (Olympic athletes) | Pathways involved with folate and amino acid biosynthesis, besides pathways involved with flavin biosynthesis and fermentation of sugar alcohols were representative in Olympic athletes. | Authors do not report significant correlations between metabolites and any individual species or pathways. |
| Liang et al. [54] | The abundance of functional categories of 28 Wushu martial were predict using PICRUSt based on closed reference OTUs. | Microbial gene functions related to histidine metabolism, chloroalkane and chloroalkene degradation and carbohydrate metabolism, were higher in elite Wushu martial artists. | Authors do not report significant correlations between microbial gene functions and any individual species or pathways |
| Allen et al. [58] | Concentrations of SCFAs was determined by gas chromatography before and after 6 weeks of aerobic training protocol. Functional Gene Quantification was assessed by qPCR | Aerobic exercise increased fecal concentrations of acetate, propionate and butyrate. Abundance of the butyrate-regulating gene BCoAT and the propionate-regulating gene mmdA was also observed after six weeks of training protocol. Results were dependent on BMI. | As a result of exercise program abundance of bacterial genera *Roseburia* spp., *Lachnospira* spp., *Clostridiales* spp., *Faecalibacterium* spp., and *f Lachnospiraceae unclass.*, positively correlated with changes in butyrate and abundance of functional genes (BCoAT) whereas changes in *Bacteroides* spp. and *Rikenella* spp. negatively correlated with changes in butyrate and/or BCoAT gene. |
| Munukka et al. [59] | Metagenome analysis was performed after exercise training in 17 sedentary middle age women. | The metagenomics assessment of the functional genes revealed no changes in the major pathways after the exercise period enrichment of pathways were not altered | N/A |
| Zhao et al. [62] | Metabolites concentration in feces was measured by LC-MS in amateur runners before and after a half marathon. KEGG ID were used for KEGG pathway enrichment analysis and PICRUSt analysis was used to predict COG functions. | Diverse metabolites from the pentose phosphate pathway, pyrimidine metabolism, and phenylalanine, tyrosine, and tryptophan biosynthesis were significantly enriched. Three metabolites in the pyrimidine metabolism pathway was decreased after running the half marathon. | Functions of gut microbiota that significantly changed after race: ↑Cell motility function ↓Energy production and conversion Pathways associated with cell motility: ↑ Flagellar assembly ↑ Bacterial chemotaxis ↑Oxidative phosphorylation related to Energy production and conversion function. |

*(Continued)*

**Table 3.** (Continued)

| Reference | Metabolic approach used across studies | Outcomes related to functional changes | Possible pathways related to gut microbiota function and composition |
|---|---|---|---|
| Scheiman et al. [55] | Meta omics analysis | Identification of *Veillonella* (genus) as a functional member of gut microbiome related to athletes. | After numerous analysis researchers concluded that the metabolic pathway *Veillonella* species utilize for lactate metabolism is enriched specially in sportive people. This symbiotic member of gut microbiota could metabolize the systemic lactate resulting from muscle activity during exercise. |

Arrows denote an increase (↑) or decrease (↓). Significant results are shown (p < 0.05). **mWNGS** = whole metagenome shotgun; **KEGG** = Kyoto Encyclopedia of Genes and Genomes; **1H-NMR** = Proton nuclear magnetic resonance; **UPLC–MS** = Ultra performance liquid chromatography–mass spectrometry; **BMI** = Body mass index; **LC–MS** = Liquid chromatography–mass spectrometry; **COG** = Clusters of Orthologous Groups; **CRF** = Cardio respiratory fitness; **HI** = High; **AVG** = Average.

an increased in butyrate concentration and butyrate producers in different studies, this short chain fatty acid is considered beneficial [66] to gut health since it is a major energy source for enterocytes and plays a key role in the maintenance of epithelial homeostasis [66].

Similar findings have been reported for *Akkermansia muciniphila*, an abundant intestinal bacterium from the *Verrucomicrobia* phylum that has been related to healthy intestinal microbiota due to its capacity to colonize the mucosal layer and improve host metabolic immune responses by increasing mucus thickness [67, 68]. *A. muciniphila* also plays a key role in metabolic activity, leading to the production of beneficial SCFAs for hosts and other microbiota members [67–69]. Another common bacterium with increased abundance in active individuals is *Faecalibacterium prausnitzii*. This *Firmicutes* member is associated with immunomodulatory properties, leading to reduced levels of IL-2, the production of Interferon-gamma, and increased secretion of the anti-inflammatory cytokine IL-10 [70]. Evidence suggests a link between *F. prausnitzii* depletion and the onset of inflammatory bowel disease and Crohn's disease. Additionally, several studies also suggest that *F. prausnitzii* produces butyrate [66, 71]. Recently, the relationship between depletion of this gut microbiota member and the presence of sarcopenia has been explored in a small groups of older adults [72], the shotgun metagenomic sequencing approach used for this study allowed to identify a depletion in microbial genes involved in diverse metabolic pathways mainly SCFA synthesis, carotenoid and isoflavone biotransformation and amino acid interconversion in sarcopenic adults when compared to non-sarcopenic counterparts [72]. Although it is not possible to determinate if these metabolic pathways are specifically related to the abundance of *F. prausnitzii* or another gut microbial member, it is of interest that numerous studies involving active people report an increase of this *Firmicutes* member, future studies in this field should aim to study the relationship between the metabolic function of *F. prausnitzii* and physical activity in different age groups [73, 74].

Another microorganism with notable abundance among sportspeople is *Eubacterium hallii*, a commensal bacterium species that contributes to the formation of butyrate from glucose fermentation [75]; *E. halli* has trophic interactions with other bacteria from the *Bifidobacterium* family, which is beneficial for host metabolism [75, 76] and is capable of metabolizing 3-hydroxypropionaldehyde—an important compound in reuterin system, which highlights the importance of this strict anaerobe since it might be capable to catalyze the transformation of dietary cancerogenic PhIP to non-carcenogenic PhIP-M1 and thus exhibiting a protective function in the colon [75, 77].

Another interesting member of gut microbiota reported in active subjects is *Gordonibacter massiliensis* [53], a Gram-positive, motile non-spore-forming and obligate anaerobic

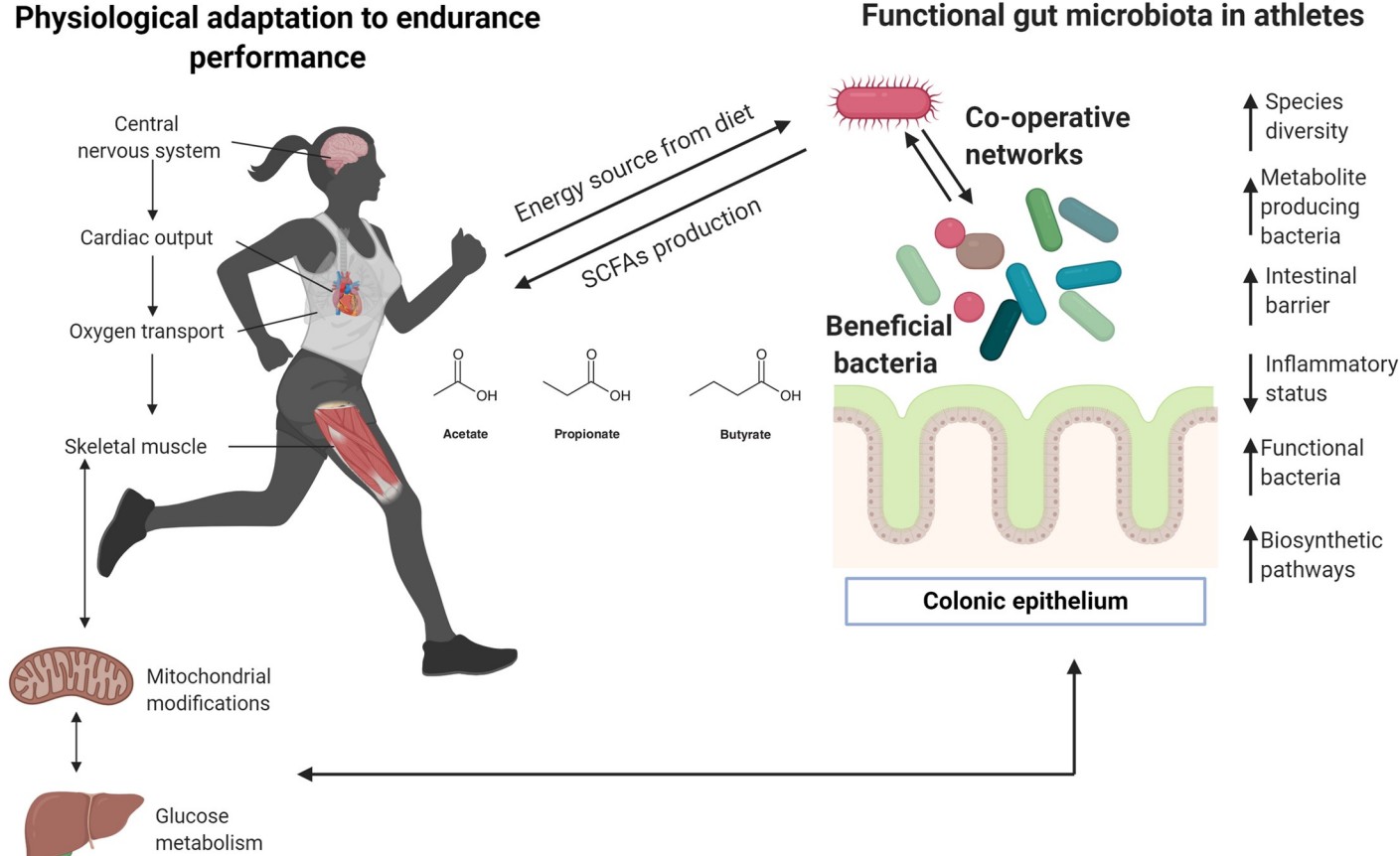

**Fig 3. Insights about meta-community adaptation among gut microbiota in athletes.** Most human physiological systems adapt to performance endurance, especially in elite and competitive athletes. Clear differences between physically inactive and athletic individuals have been described previously, particularly regarding cardiorespiratory adaptation given that endurance activities require major oxygen capacity to transport oxygen to different organs, including muscles and the liver. Efficient glucose and fatty acids metabolism are required to provide substrates that are finally transformed into energy by mitochondria. Recent research indicates that unique gut microbiota may be present in elite sportspeople, and special and unique bacteria can positively impact the host, providing substrates from the diet. Here, it is proposed that modifications of the gut microbiota ecosystem need to create co-operative networks to improve metabolic functions, particularly the production of biometabolites that can be used for the host (in this case, during highly demanding performance activities). Arrows denote an increase (↑) or decrease (↓). Figure created with Biorender.com.

coccobacillus [78]. A similar member of the *Eggerthellacea* family (*G. urolithinfaciens)* can metabolize polyphenols from the diet into urolithin [79]. This bioavailable metabolite has been tested in cell lines and murine models as a new regulator of muscle trophism, as well as being implicated in androgenic pathways possibly via AMP-activated protein kinases to enhance (a) growth of myotubes and (b) protein synthesis with ulterior muscular hypertrophy [37]. Several biological activities have been related to urolithin B, including anti-inflammatory and antioxidant properties that play a protective role against neuroinflammation [44, 46]. It is not yet clear if *G. massilensis* can promote these effects via physical activity, and further research is needed to clarify this topic [80].

Some species detected in healthy and athletic humans are widely considered as candidate probiotics due to their beneficial effects. *Bifidobacterium adolescentis* exhibits strong resistant starch degrading activity [81]. *B. longum* has been extensively researched for its neuroenteric properties including stimulation of pathways between the gut and brain via the vagus nerve; normalization of anxiety-like behavior and hippocampal brain-derived neurotrophic factor in mice with infectious colitis; and modulation of resting neural activity, reduction of mental

fatigue, and improving activation of brain coping centers to counter-regulate negative emotions [82, 83]. *Roseburia hominis* is another flagellated gut anaerobic bacterium found in women with an active lifestyle and has been reported to exhibit immunomodulatory properties that could help treat gut inflammation, with beneficial properties for the intestinal barrier [84, 85]. *R. hominis* has been considered as a potential probiotic treatment [84, 85].

At the genus level, several species from *Parabacteroides* have been associated with the amelioration of obesity and the production of succinate and secondary bile acids [86, 87]. Among the *Firmicutes* phylum, the *Phascolarctobacterium* genus can produce SFCAs like acetate and propionate, and *Phascolarctobacterium* species have been reported in healthy and active individuals [88], with positive relationships with insulin sensitivity and secretion [89]. Although previous studies report diverse outcomes in subjects undertaking exercise programs, these differences could be attributed to differences in exposure times, which varied from 6-week to 6-month interventions. Besides, BMI and physical inactivity could be determinant factors for gut microbiota responses to exercise. The main type of exercise used in research thus far is endurance training (aerobic), in which the oxidative system is the most important pathway for energy production. Once a physically inactive subject initiates an aerobic training program, it typically takes between 4 to 6 weeks to exhibit acute cardiovascular adaptation to exercise depending on cardiac output and the capacity of active muscle to extract oxygen from arterial blood [90, 91]. Fig 2 represents a possible progressive effect exerted by exercise on gut microbiota once a physically inactive subject begins endurance training and enhances their CRF.

Exercise could also contribute to communication pathways between microbiota members, which constitute metacommunities that need to function in symbiotic environments (Table 3). The outcomes analyzed in this review suggest the importance of symbiotic bacteria, which can influence and enhance the function of other members of the gut ecosystem. The application of omics approaches may be crucial to identify other microorganisms with potential functional impact, as well as to verify possible relationships with systems and organelles proposed in past reviews [92]—especially with the mitochondria [93]. Notwithstanding the exploration of axes between gut microbiota and organs are beyond the scope of this review, studies included in this section allow identifying a relationship between gut, muscle, and brain that could explain a beneficial role physical activity on gut microbiota. This hypothesis has been proposed in past reviews where authors aim to explore physical activity as a key factor between neurodegenerative diseases and gut microbiota [94], and exercise as a modulator of intestinal microbiome [95]. Since it is not possible to draw firm conclusions about these interactions it is important to highlight the need to explore different sequencing methods and the incorporation of multi omics analysis to understand the metabolic adaptation of gut microbiota to exercise. Thus, some determinant factors such as training level, intensity and frequency of exercise might be relevant to elucidate the adaptative response of gut microbiome to exercise [95], therefore they must be taken into account in future studies.

Athletes typically display exceptional CRF [45, 96]. VO2max in competitive athletes can be up to twice as high as that of sedentary subjects, leading to a higher capacity for oxidative metabolism in muscles, better neural connections, and improved general metabolism [17] (Fig 3 and Table 3). The capacity to perform endurance activities, such as a marathon, depends on many factors, particularly muscle-buffering capacity and lactate metabolism [96]. Findings from longitudinal studies may help to understand the link between athletic gut microbiota and high-performance activities. Identification and isolation of *Veillonella* in endurance athletes suggests the communication between gut microbiota and muscle mediated by physical activity and the potential impact of anaerobic bacteria on propionate production via the Cori cycle should be further examined [55]. As shown in Tables 1 and 2, descriptive studies on the microbiome composition of high-performance athletes are limited. Although some relationships

between competition level and diversity/abundance of taxa have been established [48], it is important to examine information relevant to the phenotype as well as physiological adaptations which can vary substantially from one sports discipline to another.

Diet composition should not be overlooked and more detailed information should be gathered since the use of food frequency questionnaires gives a qualitative overview of food ingestion [97]. The use of dietary assessments methods like food diaries [98] can aid the quantitative and qualitative exploration of macro and micronutrients consumption [99]; it would also be relevant to determine the preliminary use of probiotics and prebiotics since its consumption has become recurrent in athletic population especially encouraged by the reported regenerative and immunologic benefits of *Lactobacillus*, *Bifidobacterium*, and *Bacillus* genera [26].

## Limitations

This review attempts to summarize the main outcomes related to diversity and relative abundance across human studies involving different PAA. Although numerous studies have provided approaches for elucidating possible mechanisms, the effects of individual characteristics and the numerous factors that can influence the composition of the microbiome are important to consider. All studies included in this review were performed in relatively developed high-income countries. Although some reports have included the Hispanic population [50, 51], there is an important data gap from developing regions, particularly South America and Africa, which display substantial variations in the diet [100]. Similarly, the measurement of inflammatory and metabolic markers could provide useful information regarding the metabolism and physical condition of the host. The consumption of supplements and ergogenic aids from athletes should also be described in this field.

We also suggest that future randomized and non-randomized studies should try to implement direct methods like accelerometry to measure variables related to physical activity instead of auto-reported methods. Similarly, direct oxygen consumption measures, energy expenditure, and heart rate variability could help elucidate the systematic response of gut microbiota to exercise.

## Conclusions

Our review aims to explore the possibility that exercise promotes the abundance of intestinal bacteria that could drive beneficial metabolic changes to human hosts (Figs 2 and 3). Further research is needed in the form of controlled clinical trials including different types of exercise (i.e. endurance and high-intensity training), distinct age groups, larger samples and incorporation of multi-omics approaches, as well as detailed information about diet and others lifestyle factors like sleep and circadian rhythm.

Few studies have expanded to other types of microorganisms, such as viruses [101] and archaea [48], while the role of parasites and fungi has not yet been identified. Exploration of other groups may help to comprehend the role of the gut microbiome on exercise adaptation, particularly muscle growth and biochemical signaling. One recent study reported the use of adeno-associated virus 9 (AAV9) as a vector to deliver the protein follistatin to improve muscle performance and mitigate the severity of osteoarthritis sequelae, including inflammation and obesity in mice [102]. Although this is a gene therapy approach, the possibility that viruses could mediate signaling pathways in short-to-long-term adaptation to exercise should not be neglected. The meta-community in the intestinal environment emphasizes the potential connections between bacteria, fungi, parasites, and viruses, which could directly influence the response to exercise [103, 104].

To date, it is not yet possible to draw firm conclusions about whether the metabolic activity of bacteria can impact various tissues via metabolites production during exercise in healthy humans. Nevertheless, the results reported by Scheiman et al. [55] appear to be significant, identifying a stable relationship between possible efficient microbiota members and the production of propionate, impacting lactate production, at least in the colonic lumen.

Future research examining the microbiota-exercise association should also aim to describe aspects of the lifestyle as well as possible, including diet, the level of training, and sedentary behavior.

## Supporting information

**S1 Checklist. PRISMA 2009 checklist.**
(DOC)

**S1 Table. ROBINS-I risk of bias assessment summary: Review authors' judgements about each methodological quality item for each included study in this review.**
(DOCX)

## Acknowledgments

We thank Benjamin Knight, MSc., from Edanz Group (https://en-author-services.edanzgroup.com/) for editing a draft of this manuscript.

## Author Contributions

**Conceptualization:** Viviana Aya, Alberto Flórez, Juan David Ramírez.

**Data curation:** Viviana Aya.

**Formal analysis:** Viviana Aya, Luis Perez, Juan David Ramírez.

**Investigation:** Viviana Aya, Juan David Ramírez.

**Methodology:** Viviana Aya, Juan David Ramírez.

**Resources:** Juan David Ramírez.

**Supervision:** Alberto Flórez, Luis Perez, Juan David Ramírez.

**Validation:** Viviana Aya, Alberto Flórez, Luis Perez, Juan David Ramírez.

**Visualization:** Juan David Ramírez.

**Writing – original draft:** Viviana Aya.

**Writing – review & editing:** Alberto Flórez, Luis Perez, Juan David Ramírez.

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
