## [Decision Letter · Decision Letter 0]

23 Dec 2020

PONE-D-20-35235

ASSOCIATION BETWEEN PHYSICAL ACTIVITY AND CHANGES IN INTESTINAL MICROBIOTA COMPOSITION: A SYSTEMATIC REVIEW

PLOS ONE

Dear Dr. Ramírez,

Thank you for submitting your manuscript to PLOS ONE. After careful consideration, we feel that it has merit but does not fully meet PLOS ONE’s publication criteria as it currently stands. Therefore, we invite you to submit a revised version of the manuscript that addresses the points raised during the review process.

We look forward to receiving your revised manuscript.

Kind regards,

Jane Foster, PhD

Academic Editor

PLOS ONE

Journal Requirements:

2. Please include captions for your Supporting Information files at the end of your manuscript, and update any in-text citations to match accordingly. Please see our Supporting Information guidelines for more information: http://journals.plos.org/plosone/s/supporting-information

Reviewers' comments:

Reviewer's Responses to Questions

**Comments to the Author**

1. Is the manuscript technically sound, and do the data support the conclusions?

Reviewer #1: Yes

Reviewer #2: Yes

2. Has the statistical analysis been performed appropriately and rigorously? 

Reviewer #1: Yes

Reviewer #2: Yes

3. Have the authors made all data underlying the findings in their manuscript fully available?

Reviewer #1: Yes

Reviewer #2: Yes

4. Is the manuscript presented in an intelligible fashion and written in standard English?

Reviewer #1: Yes

Reviewer #2: Yes

5. Review Comments to the Author

Reviewer #1: Dear Author

the topic of the manuscript is really relevant and actual.

In fact, the relationship between microbiota and physical activity could be the key factor

between physical activity and neurodegenerative diseases, creating an "gut-muscle-brain" axis.

Recently it as been published an article:

Ticinesi A, Mancabelli L, Tagliaferri S, Nouvenne A, Milani C, Del Rio D, Lauretani F, Maggio MG, Ventura M, Meschi T.

The Gut-Muscle Axis in Older Subjects with Low Muscle Mass and Performance: A Proof of Concept Study Exploring Fecal Microbiota Composition and Function with Shotgun Metagenomics Sequencing.

Int J Mol Sci. 2020 Nov 25;21(23):E8946. doi: 10.3390/ijms21238946

that could be used for explaining this relatioship.

And also this other article could be used for close this loop:

Ticinesi A, Lauretani F, Tana C, Nouvenne A, Ridolo E, Meschi T.

Exercise and immune system as modulators of intestinal microbiome: implications for the gut-muscle axis hypothesis.

Exerc Immunol Rev. 2019;25:84-95.

and finally:

Ticinesi A, Tana C, Nouvenne A, Prati B, Lauretani F, Meschi T.

Gut microbiota, cognitive frailty and dementia in older individuals: a systematic review.

Clin Interv Aging. 2018 Aug 29;13:1497-1511.

Reviewer #2: In this study, Aya et.al. present a systematic review of literature describing the association between physical activity and intestinal microbiota in human subjects. From the studies they analyzed, the authors have provided a comprehensive overview of how exercise affects intestinal microbiota in humans. The review is well written, succinct, and gives a good overview of the studies considered. However, I have the following comments:

1. There have been recent reviews on the topic of exercise and gut microbiome composition in athletes vs non-athletes. Almost all the reviews describe the changes of gut microbiome in terms of the composition. I urge the authors to include a table like the other tables that provides a description of the downstream functional changes associated with the compositional information. The discussion section provides a good template for this.

2. Minor comment: There are minor typos in the document e.g. TD2 instead of T2D for Type II Diabetes.

6. PLOS authors have the option to publish the peer review history of their article (what does this mean?). If published, this will include your full peer review and any attached files.

Reviewer #1: No

Reviewer #2: No

---

## [Author Response · Author response to Decision Letter 0]

14 Jan 2021

January 14th, 2021

Bogotá D.C., Colombia

Jane Foster, PhD

Academic Editor

PLOS ONE

Dear Dr. Foster,

First of all, we want to thank your consideration with our manuscript. Please find below as requested the specific responses to the comments and suggestions raised by the reviewers.

COMMENTS FROM REVIEWER # 1

COMMENT: The topic of the manuscript is relevant and actual. In fact, the relationship between microbiota and physical activity could be the key factor between physical activity and neurodegenerative diseases, creating a "gut-muscle-brain" axis.

Recently it has been published an article: 

• Ticinesi A, Mancabelli L, Tagliaferri S, Nouvenne A, Milani C, Del Rio D, Lauretani F, Maggio MG, Ventura M, Meschi T. 

The Gut-Muscle Axis in Older Subjects with Low Muscle Mass and Performance: A Proof of Concept Study Exploring Fecal Microbiota Composition and Function with Shotgun Metagenomics Sequencing. Int J Mol Sci. 2020 Nov 25;21(23):E8946. doi: 10.3390/ijms21238946

That could be used for explaining this relationship. 

RESPONSE: We agree with the inclusion of this recent study. A brief reference to this article can be found in “discussion section” specifically between lines 319 to 329, reference 73. 

COMMENT: And also this other article could be used for close this loop:

Ticinesi A, Lauretani F, Tana C, Nouvenne A, Ridolo E, Meschi T. Exercise and immune system as modulators of intestinal microbiome: implications for the gut-muscle axis hypothesis. Exerc Immunol Rev. 2019;25:84-95.

RESPONSE: We agree with the inclusion of this review. A brief reference to the article can be found in “discussion section” specifically between lines 384 to 395, reference 98. 

COMMENT: and finally:

Ticinesi A, Tana C, Nouvenne A, Prati B, Lauretani F, Meschi T. Gut microbiota, cognitive frailty and dementia in older individuals: a systematic review. Clin Interv Aging. 2018 Aug 29;13:1497-1511.

RESPONSE: We agree with the inclusion of this review. A brief reference of the article can be found in “discussion section” specifically between lines 384 to 395, reference 97. 

COMMENTS FROM REVIEWER #2

In this study, Aya et.al. present a systematic review of literature describing the association between physical activity and intestinal microbiota in human subjects. From the studies they analyzed, the authors have provided a comprehensive overview of how exercise affects intestinal microbiota in humans. The review is well written, succinct, and gives a good overview of the studies considered. However, I have the following comments:

COMMENT: There have been recent reviews on the topic of exercise and gut microbiome composition in athletes vs non-athletes. Almost all the reviews describe the changes of gut microbiome in terms of the composition. I urge the authors to include a table like the other tables that provides a description of the downstream functional changes associated with the compositional information. The discussion section provides a good template for this.

RESPONSE: We certainly agree with the inclusion of a new table named “Description of possible functional changes associate with gut microbiota in physical activity and sports activities” (Lines 284 to 289) 

Additionally, some lines were included in the text to introduce and explain the table, these can be found in the lines: 266 – 267; 304 – 305; 379; 400. 

COMMENT: There are minor typos in the document e.g. TD2 instead of T2D for Type II Diabetes.

RESPONSE: Typo mistakes were corrected in lines 64 and 74.

---

## [Editor Report · Decision Letter 1]

1 Feb 2021

ASSOCIATION BETWEEN PHYSICAL ACTIVITY AND CHANGES IN INTESTINAL MICROBIOTA COMPOSITION: A SYSTEMATIC REVIEW

PONE-D-20-35235R1

Dear Dr. Ramírez,

We’re pleased to inform you that your manuscript has been judged scientifically suitable for publication and will be formally accepted for publication once it meets all outstanding technical requirements.

Kind regards,

Jane Foster, PhD

Academic Editor

PLOS ONE
---

## [Editor Report · Acceptance letter]

3 Feb 2021

PONE-D-20-35235R1 

Association Between Physical Activity and Changes in Intestinal Microbiota Composition: A Systematic Review 

Dear Dr. Ramírez:

I'm pleased to inform you that your manuscript has been deemed suitable for publication in PLOS ONE. Congratulations! Your manuscript is now with our production department. 

Kind regards, 

on behalf of

Dr. Jane Foster 

Academic Editor

PLOS ONE